# On the Nanoscale Mapping of the Mechanical and Piezoelectric Properties of Poly (L-Lactic Acid) Electrospun Nanofibers

**Nguyen Thai Cuong [1,2], Sophie Barrau [2], Malo Dufay [2], Nicolas Tabary [2], Antonio Da Costa [3], Anthony Ferri [3], Roberto Lazzaroni [1], Jean-Marie Raquez [4] and Philippe Leclère [1,*]**

[1] Laboratory for Chemistry of Novel Material, Center for Innovation and Research in Materials and Polymers (CIRMAP), University of Mons (UMONS), B-7000 Mons, Belgium; ThaiCuong.NGUYEN@umons.ac.be (C.N.T.); roberto.lazzaroni@umons.ac.be (R.L.)

[2] Université Lille, Sciences et Technologies, CNRS, ENSCL, INRA, UMR 8207, Unité Matériaux Et Transformations (UMET), F-59655 Villeneuve D'Ascq, France; sophie.barrau@univ-lille.fr (S.B.); malo.dufay@univ-lille.fr (M.D.); nicolas.tabary@univ-lille.fr (N.T.)

[3] Université Artois, CNRS, Centrale Lille, ENSCL, Université Lille, UMR 8181, Unité de Catalyse et Chimie Du Solide (UCCS), F-62300 Lens, France; antonio.dacosta@univ-artois.fr (A.D.C.); anthony.ferri@univ-artois.fr (A.F.)

[4] Laboratory of Polymeric and Composite Materials, Center for Innovation and Research in Materials and Polymers (CIRMAP), University of Mons (UMONS), B-7000 Mons, Belgium; jean-marie.raquez@umons.ac.be

\* Correspondence: philippe.leclere@umons.ac.be

**Abstract:** The effect of the post-annealing process on different properties of poly (L-lactic acid) (PLLA) nanofibers has been investigated in view of their use in energy-harvesting devices. Polymeric PLLA nanofibers were prepared by using electrospinning and then were thermally treated above their glass transition. A detailed comparison between as-spun (amorphous) and annealed (semi-crystalline) samples was performed in terms of the crystallinity, morphology and mechanical as well as piezoelectric properties using a multi-technique approach combining DSC, XRD, FTIR, and AFM measurements. A significant increase in the crystallinity of PLLA nanofibers has been observed after the post-annealing process, together with a major improvement of the mechanical and piezoelectric properties.

**Keywords:** electrospinning; nanofibers; peakforce quantitative nanomechanical property mapping; piezoresponse force microscopy; intermodulation AFM; contact mechanics; poly (L-lactic acid)

## 1. Introduction

Recently, materials processed as nanofibers have attracted extensive interest for their wide applications in various fields such as drug delivery [1], tissue engineering scaffolds [2,3], air filtration, water treatment [4], sensors [5], body thermoregulation [6,7] as well as energy conversion and storage [8]. Together with nanowires, nanotubes and nanorods, nanofibers belong to the group of one-dimensional (1D) nanostructured materials displaying unique physical and chemical characteristics. Fabrication of nanofibrous materials has been practiced by a wide range of methods: template-assisted synthesis [9], physical and chemical vapor deposition [10], self-assembly [11], centrifugal spinning [12], thermal-induced phase separation [13] and electrospinning. Electrospinning has the advantages of being a simple, scalable and low-cost technique. This method can be applied to a broad range of materials including inorganic materials [14], natural and synthetic polymers [15] and composite nanomaterials [16] with final products having large surface areas and a higher aspect ratio. Furthermore, by controlling

process parameters including, starting materials, applied voltage, feed rate, working distance or collector type, various morphologies and network structures such as porous [17], core-shell [18], hollow [19] or aligned nanofibers [20] can be generated.

Poly (L-lactic acid) (PLLA) is a polymer that has biocompatible and biodegradable characteristics. Being derived from biomass, it is widely used as a commercial product and is found in various applications. Its morphology and mechanical properties depend on many factors such as the polymer molecular weight, the processing methods, leading to either amorphous or semi-crystalline PLLA [21,22]. In particular, the nanofibers generated by the electrospinning process can be either amorphous or poorly crystalline due to the fast evaporation of the solvent. In this case, the products can show poor performance and, therefore, require some post-treatment processes such as thermal treatment (or annealing) above the glass transition. As a result, the crystallinity, the morphology, the mechanical properties and the piezoelectricity of the nanofibers are greatly improved [23–26].

Atomic force microscopy (AFM) is a powerful technique for characterizing a variety of properties at nanoscales. It is a particularly suitable tool to investigate nanomechanical properties of nanometer-sized materials, including nanofibrous materials. Following the rapid advance of technology, different surface property mapping, AFM-based techniques have been developed with more precise control of the force, higher speed, larger data acquisition as well as the ability to distinguish viscoelasticity. Among these approaches, peakforce QNM [27] (PFQNM) and intermodulation AFM [28] (ImAFM) are powerful, commercially available techniques. Both techniques have been successfully deployed to study nanomechanical responses of different materials [29,30]. The combination of those techniques will give us a full picture of the elastic and viscoelastic properties of any given material. Piezoresponse force microscopy (PFM) is an AFM-based tool that focuses on characterizing local ferroelectric and piezoelectric properties [31,32]. Some authors like Sencadas et al. [33] and Sultana et al. [34] used PFM to show that PLLA electrospun nanofibers exhibit ferroelectricity and piezoelectricity in the direction perpendicular to the fiber main axis, i.e., the existence of a piezoelectric coefficient, $d_{33}$. Their research provides promising results for the application of PLLA electrospun nanofibers in the field of energy storage devices and nanogenerators. However, a more-complete picture is required. Indeed, research concerning the effect of crystallinity, as well as morphologies on nanomechanical and piezoelectric properties of electrospun nanofibers is still missing.

In the present work, the morphology and the crystallinity of electrospun PLLA nanofibers have been controlled by thermal post-treatment, leading to improved mechanical and piezoelectrical properties. A complete investigation of pristine and annealed nanofibers was performed using a multi-technique approach. On one hand, classical analytical techniques such as XRD, FTIR, and DSC were used to examine changes in crystallinity and identify the polymer crystalline phase in the annealed nanofibers. On the other hand, the implementation of AFM-based methods unveiled interesting results related to local mechanical and piezoelectric properties. Special attention has been paid to the influence of the amorphous or semi-crystalline nanofibers on the morphology and properties at the nanoscale.

## 2. Materials and Methods

### 2.1. Materials

Poly (L-lactic acid) (PLLA, trade name Luminy PLA L130 with ≥99% (L-isomer) supplied by Total Corbion (Gorinchem, The Netherlands), dichloromethane (DCM) (from Sigma–Aldrich, Darmstadt, Germany), acetonitrile (from Sigma–Aldrich, Darmstadt, Germany), aluminum foils (LABOR, Schaffhausen, Switzerland), glass substrates (from Sigma–Aldrich, Darmstadt, Germany); indium tin oxide-coated glass substrates (from Naranjo Substrates, Groningen, The Netherlands).

### 2.2. PLLA Electrospun Nanofibers

PLLA pellets were dissolved in a mixture of DCM and acetonitrile (1/1 *v/v*) to achieve a solution with a polymer concentration of 6 wt%. The solution was continuously stirred by a magnetic stirrer

until the pellets were completely dissolved. The polymer solution was then placed in a commercial plastic syringe (10 mL) fitted with a steel needle with 500 μm inner diameter. The electrospinning of the PLLA solution was performed using an automated electrospinning instrument, Fluidnatek LE-500 instrument (Bioinicia, Valencia, Spain). During the electrospinning process, the voltage was kept at 13.5 kV for the needle and −10 kV for the collector. The solution was fed into the needle tip at a constant rate of 5 mL h$^{-1}$. The electrospun fibers were collected in a static collecting plate placed a distance of 20 cm away from the needle. The electrospinning was conducted at 25 °C with a relative humidity of 30%. For characterization, three types of substrates were used: aluminum sheets for structural analysis, glass substrates for the investigation of the morphology and mechanical properties at the nanoscale and conductive substrates such as indium tin oxide (ITO)-coated glass substrates for the investigation of the piezoelectric properties.

### 2.3. Annealing of PLLA Nanofibers

The electrospun PLLA fibers were put into an oven and kept for 2 h at three different temperatures: 90, 100 or 120 °C. The annealed samples were compared with the untreated samples to investigate the effect of annealing process.

### 2.4. Characterization Methods

#### 2.4.1. Thermal Analysis

Differential scanning calorimetry (DSC) was used to study the thermal properties and phase transitions of the electrospun PLLA fibers by using a DSC Q200 (TA instrument). The glass transition temperature ($T_g$), cold crystallization temperature ($T_c$) and melting temperature ($T_m$) of the samples were measured. As-prepared and annealed electrospun fibers deposited on aluminum sheets were first peeled off from the metallic substrate and a sufficient number of samples were put inside an aluminum pan (5–7 mg). The samples were then heated from 20 to 200 °C with a heating rate of 10 °C min$^{-1}$. The crystalline content ($X_c$) of the sample can be estimated quantitatively, as calculated by the following equation:

$$X_c = \frac{\Delta H_m - \Delta H_c}{\Delta H_m^0} \tag{1}$$

where $\Delta H_m$ and $\Delta H_c$ are the enthalpies of melting and cold crystallization determined from the DSC thermogram, respectively. $\Delta H_m$ is the enthalpy of melting for a PLA single crystal which is 93.7 J/g [35].

#### 2.4.2. Fourier Transform Infrared Spectroscopy (FTIR) Investigation

Fourier transform infrared spectroscopy (FTIR) analysis were carried out on a Perkin Elmer Spectrum 100 spectrometer (PerkinElmer, Zaventem, Belgium) in ATR (attenuated total reflection) mode. Spectra were obtained after 16 accumulations with a resolution of 4 cm$^{-1}$.

#### 2.4.3. Wide-Angle X-ray Scattering (WAXS) Study

The structure of the as-prepared and annealed electrospun fibers was investigated by Wide-angle X-ray scattering (WAXS) on a Xeuss 2.0 (Xenocs, Grenoble, France) diffractometer with a GeniX3D microsource (λ = 1.5406 Å) operating at 0.6 mA and 50 kV and a 2D Pilatus 3R 200K detector. The sample-to-detector distance was around 120 mm.

#### 2.4.4. Atomic Force Microscopy (AFM)-Based Methods

The morphology and mechanical properties of the as-prepared and annealed electrospun PLLA nanofibers were inspected by a Dimension Icon AFM instrument (Bruker Nano Inc., Santa Barbara, CA, USA). AFM Tapping mode was employed for studying the morphology of the nanofibers using silicon cantilevers (RTESPA from Bruker Nano Inc.) with a spring constant of ~42 N/m and an apex

radius of curvature ~10 nm. For the investigation of the local mechanical properties, pre-calibrated RTESPA-300-30 tips from Bruker were used in PFQNM mode. These tips had a well-controlled tip end radius of 30 nm (± 15%) and the spring constant of each tip was individually calibrated by the laser doppler vibrometer method. The information related to tip calibration was stored and can be easily retrieved by scanning over the QR code on the tip container. In this study, the tip used had a tip radius of 29 nm and a spring constant of 49.487 N/m. For PFQNM experiments, the deflection sensitivity was estimated before the measurements from the linear part of a force-distance curve obtained by ramping the tip onto a clean Sapphire surface. The peak force amplitude was 30 nm, the scan rate was 0.5 Hz and all the captured images were recorded with a resolution of $256 \times 256$ data points. A force curve was also recorded for each pixel of the image. After the calibration step was done, the same tip and parameters (peak force amplitude, scan rate and resolution) were applied to measure the contact modulus of the PLLA electrospun nanofibers. Only the maximum force (peak force) was kept at F = 30 nN to ensure enough indentation (a few nm) into the samples while still keeping the deformation in the elastic region.

ImAFM experiments were conducted with the same AFM apparatus equipped with an intermodulation lockin analyzer and ImAFM software supplied by Intermodulation Products Company (Segersta, Sweden). We used the NCHV-A cantilever (from Bruker Nano Inc.) which is similar to NCHV used in tapping mode except that the back side of this cantilever is coated with a reflective layer of aluminum. The dynamic stiffness in air of the cantilever was calculated as $k_{air}$ = 37.52 N/m by the Sader method using the expression for the hydrodynamic damping and the calibrated inverse optical lever responsibility invOLR = 56.04 nm/V. The nanoscale electroactive responses were probed by using the dual AC resonance tracking (DART) method [36] of the piezoresponse force microscopy (PFM) technique in air at room temperature with a commercial AFM microscope (Asylum Research/Oxford Instruments, MFP-3D, Santa Barbara, CA, USA). For these experiments, nanofibers were directly deposited on conductive substrates (i.e., indium-tin-oxide-coated glass) with carbon paste in-between and the nanometric tip probed individual nanofiber. Pt/Ir-coated silicon tips and cantilevers with stiffness of about 3.5 N/m (PPP-EFM, Nanosensors) were used as the conductive probe. More specifically, PFM in spectroscopic mode was used for recording remnant PFM loops (at zero DC bias) after a continuous AC signal superimposed on an intermittent DC bias voltage was applied to the fibers to promote the electromechanical response at the expense of the electrostatic contribution.

## 3. Results and Discussion

### 3.1. Crystallinity

The DSC thermograms of as-spun and annealed nanofibers are shown in Figure 1. An exothermic peak at 85 °C, related to the polymer cold crystallization, is seen in the as-spun fibers indicating that, as expected, those samples have a low crystallinity because of the fast evaporation of the solvent during the electro-spinning process. In contrast, there is no sign of a cold crystallization peak in the annealed samples, which refers to the fact that the annealing process increases the overall crystallinity. Table 1 displays the values of the characteristic temperature ($T_c$ and $T_m$), the associated enthalpies ($\Delta H_c$ and $\Delta H_m$) and $X_c$ for each sample. The crystallinity of the nanofibers increases dramatically from 2% in the as-spun sample to about 50% in the annealed sample.

Interestingly, there is no clear shifting in melting temperature or a large difference in values of $X_c$ between samples annealed at 90, 100 and 120 °C, indicating that the crystallinity content is expected to be maximized in the samples after the thermal post-treatment. The increase of the crystalline content for all annealed samples compared to the as-prepared sample confirms the positive effect of the annealing process on the crystallinity of the electrospun nanofibers. It should be noted that the samples have a glass transition ($T_g$) around 60 °C, well below the annealing temperature.

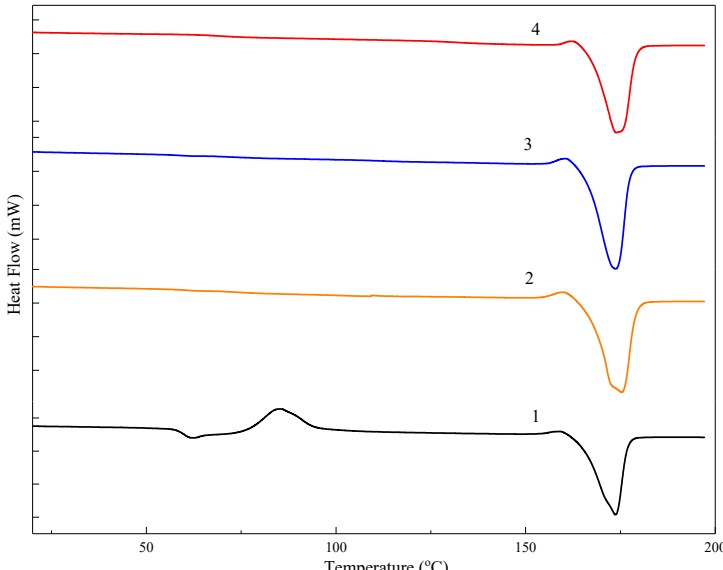

**Figure 1.** Differential scanning calorimetry (DSC) curves of electrospun poly(L-lactic acid) (PLLA) nanofibers: as-spun (1) annealed at 90 °C (2), 100 °C (3), and 120 °C (4) Exothermic upward.

**Table 1.** Values of cold crystallization temperature ($T_c$), melting temperature ($T_m$), cold crystallization enthalpy ($\Delta H_c$), melting enthalpy ($\Delta H_m$) and crystalline content ($X_c$) of as-spun (1), annealed at 90 °C (2), 100 °C (3), and 120 °C (4) nanofibers.

| Sample | $T_c$ (°C) | $\Delta H_c$ (J/g) | $T_m$ (°C) | $\Delta H_m$ (J/g) | $X_c$ (%) |
|:---:|:---:|:---:|:---:|:---:|:---:|
| (1) | 85.6 | 37.4 | 173.7 | 38.6 | 2 ± 1 |
| (2) | - | - | 175.3 | 45.8 | 49 ± 1 |
| (3) | - | - | 173.6 | 46.8 | 50 ± 1 |
| (4) | - | - | 173.9 | 46.3 | 50 ± 1 |

### 3.2. PLLA Crystal Phase

The improvement of crystallinity in annealed samples is also proved by FTIR spectroscopy. Figure 2a,b show the FTIR spectroscopy of electrospun PLLA nanofibers without and with thermal treatment in different spectral regions. According to many spectroscopic studies on the polymorphism of PLLA, it has been proven that the band at 956 cm$^{-1}$ can be assigned to the amorphous fraction, whereas the band at 921 cm$^{-1}$ is ascribed to the PLLA crystal phases, $\alpha$ and $\alpha'$ [37,38]. The appearance of the absorption band at 921 cm$^{-1}$ as well as the decrease in absorbance at 956 cm$^{-1}$ in all annealed samples with respect to those of as-spun samples are supporting evidence of the improvement of the crystalline content in the heat-treated samples. Investigation of FTIR spectroscopy also revealed information regarding crystalline phase present in the electrospun fibers. According to Kalish et al. [39], the FTIR spectrum of the $\alpha$ crystalline phase of PLA presents multiple band-splitting in the spectral region of 750–650 cm$^{-1}$, whereas the spectral patterns of the $\alpha'$ crystalline phase in the same region are much less complex. In this regard, the absence of band splitting in the 750–650 cm$^{-1}$ region for all annealed nanofibers indicates that the crystalline forms generated here are dominantly $\alpha'$. Additional evidence includes the presence of a broad band around 1750 cm$^{-1}$, corresponding to the C=O stretching vibration region, in all FTIR spectra of annealed samples. Indeed, the $\alpha'$ crystalline form is described as slightly different than the $\alpha$ form with a distorted $10_3$ helical packing chain, resulting in weaker interactions and coupling between carbonyl groups [40]. Consequently, the carbonyl band of $\alpha'$ displays a single broad band, as seen in Figure 2a.

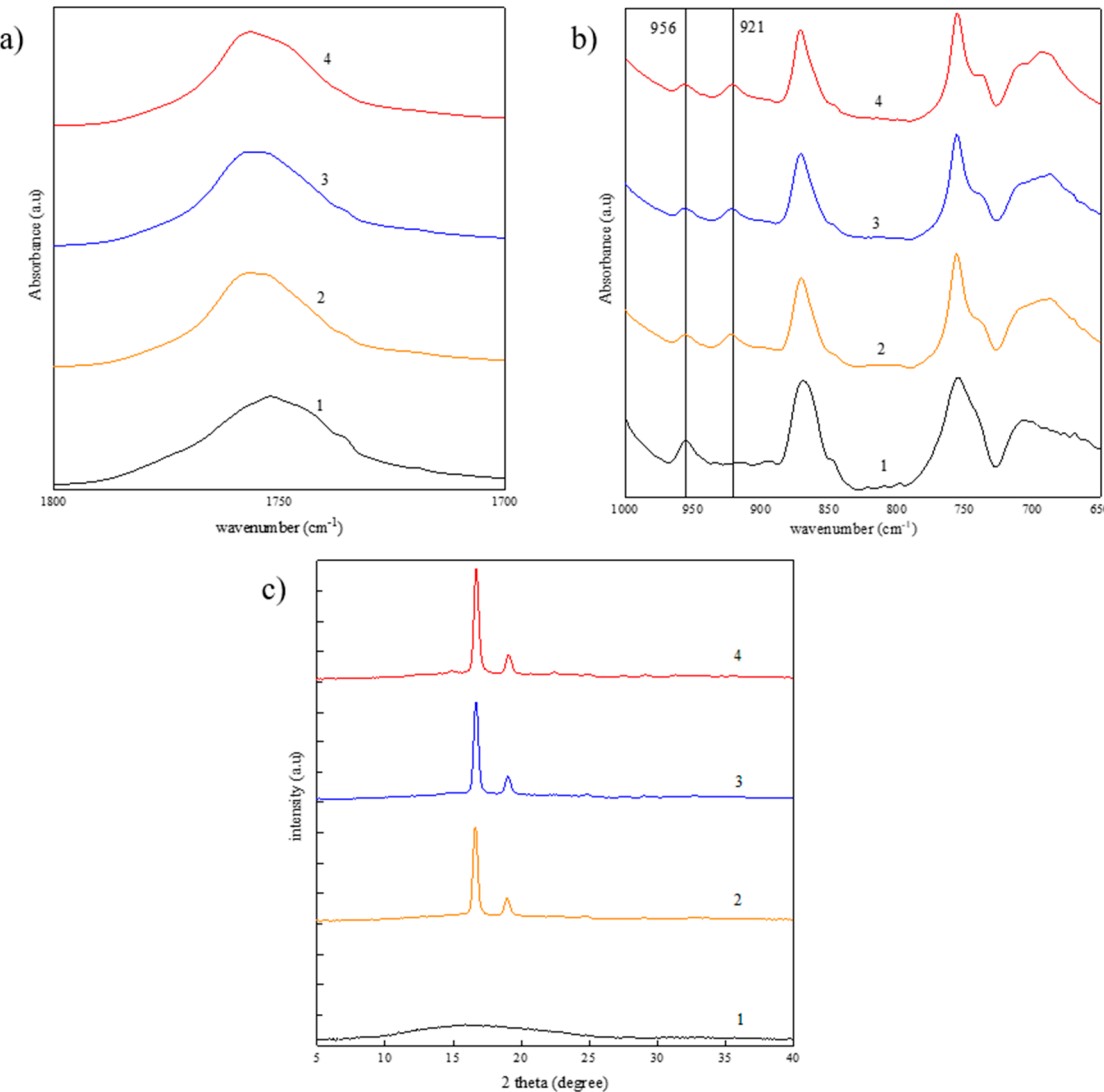

**Figure 2.** Fourier transform infrared spectroscopy (FTIR) spectra in (**a**) the C=O stretching region; and in (**b**) the region of 1000–650 cm$^{-1}$ of PLLA nanofibers. X-Ray Diffraction (XRD) patterns presented in (**c**). The numbers 1–4 refer to the as-spun sample, and annealed samples at 90, 100 and 120 °C, respectively.

In addition to FTIR spectroscopy, WAXS experiments were performed to study the crystallinity as well as the structure of electrospun PLLA fibers. Figure 2c shows the integrated intensity profile of electrospun fibers, as-prepared and heat-treated samples. Annealed samples show two sharp diffraction peaks at 2θ = 16.6° and 19.0° corresponding to the (200/110) and (203) planes respectively and attributed to the PLLA $\alpha'$-phase [41]. On the contrary, the broad signal of as-spun PLLA fiber is attributed to the intrinsic amorphous nature of the material. In short, all those results clearly show that the crystallinity of the nanofibers can be significantly improved by annealing above the polymer glass transition. Due to the fact that annealed nanofibers present the same crystalline content and only one crystalline phase ($\alpha'$) occurred in all heat-treated samples, the sample annealed at 100 °C is considered to be representative of all annealed samples and will be investigated as the annealed nanofibers in the following sections.

### 3.3. Morphologies of PLLA Nanofibers

A comparison between tapping mode AFM images of the surface topography of as-spun and annealed nanofibers is given in Figure 3. The effect of the annealing process on the topographical

features of the electrospun fibers clearly appears. The as-prepared sample shows a porous structure which most probably results from the solvent's fast evaporation during the electrospinning process [42] (Figure 3a,b). Moreover, under the stretching effect occurring during the electrospinning process, all the polymer chains are expected to orient in the direction of the fiber axis, creating elongated pores at the sample surface. Annealing above the glass transition induces a rearrangement of the surface resulting in a more textured morphology. Although the details of the texture are obscured in the height image, they clearly appear in the phase image (Figure 3b,e) as granules probably corresponding to densely packed crystallites [43]. The diameter of a single nanofiber can also be measured from the tapping height image, although this should be done with caution. The estimation made by measuring the width of the nanofibers is usually inaccurate due to the tip convolution. The average diameter of the nanofibers should instead be evaluated by measuring the distance from the substrate to the highest point of the cross-section profile. It should also be noted that, when operating in tapping mode, the AFM tip gently touches the sample surface and the force between tip and sample is kept at a minimum. Therefore, any damage to the surface can be avoided and the height values measured are the "true" heights. As observed in Figure 3c,f, the width of the nanofiber would imply a diameter of approximately 380 nm for the as-prepared sample and 320 nm for the annealed sample, whereas the height cross-section profile, which is more reliable, indicates 285 nm and 175 nm for pristine and heat-treated nanofibers, respectively.

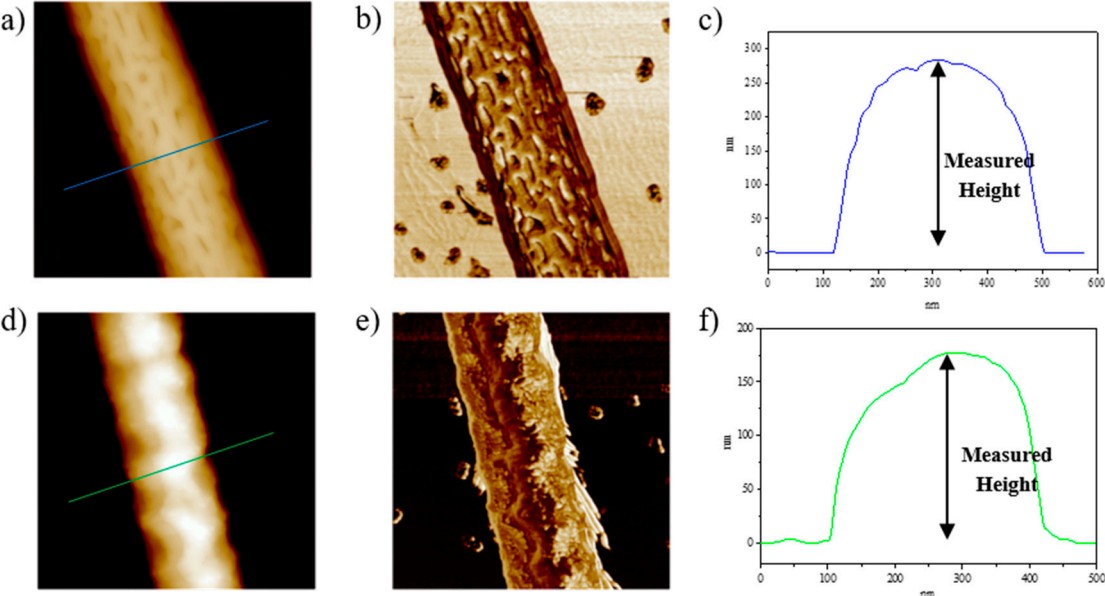

**Figure 3.** Tapping mode AFM height and phase images of as-spun PLLA fibers (**a**,**b**), and of annealed nanofibers (**d**,**e**). Cross-sectional profile of height images for as-prepared sample (**c**) and annealed sample (**f**).

The average diameter of the fibers as well as the standard deviation were then calculated from 30 different nanofibers and were found to be 210 ± 88 nm for the pristine fibers and 160 ± 44 nm for the annealed nanofibers (see some representative images in Figures S1–S3 in Supporting Information). The decrease of the diameter of the nanofibers upon annealing probably originates from the collapse of the pores and the growth and merging of crystallites, which are denser than the amorphous phase [43].

*3.4. Nanomechanical Properties of Electrospun PLLA Nanofibers*

3.4.1. Validation of the Technique

The mechanical properties of the nanofibers were first investigated using an AFM-based method called peakforce QNM (PFQNM). In this technique, the force applied to the sample is precisely

controlled provided the spring constant of the cantilever and the deflection sensitivity are known through some calibration steps [44] and a force-distance curve is recorded at each pixel during the measurement. These force curves can then be fitted with a contact model such as the Hertz model [45], the Derjaguin-Muller-Toporov (DMT) model [46] or the Johnson-Kendall-Roberts (JKR) model [47] to map the local mechanical properties of the sample. In this study, the modulus value of each nanofiber was measured over a given distance along the nanofiber axis (see Figure 4a) and the mean value of the modulus, as well as the standard deviation, can consequently be calculated.

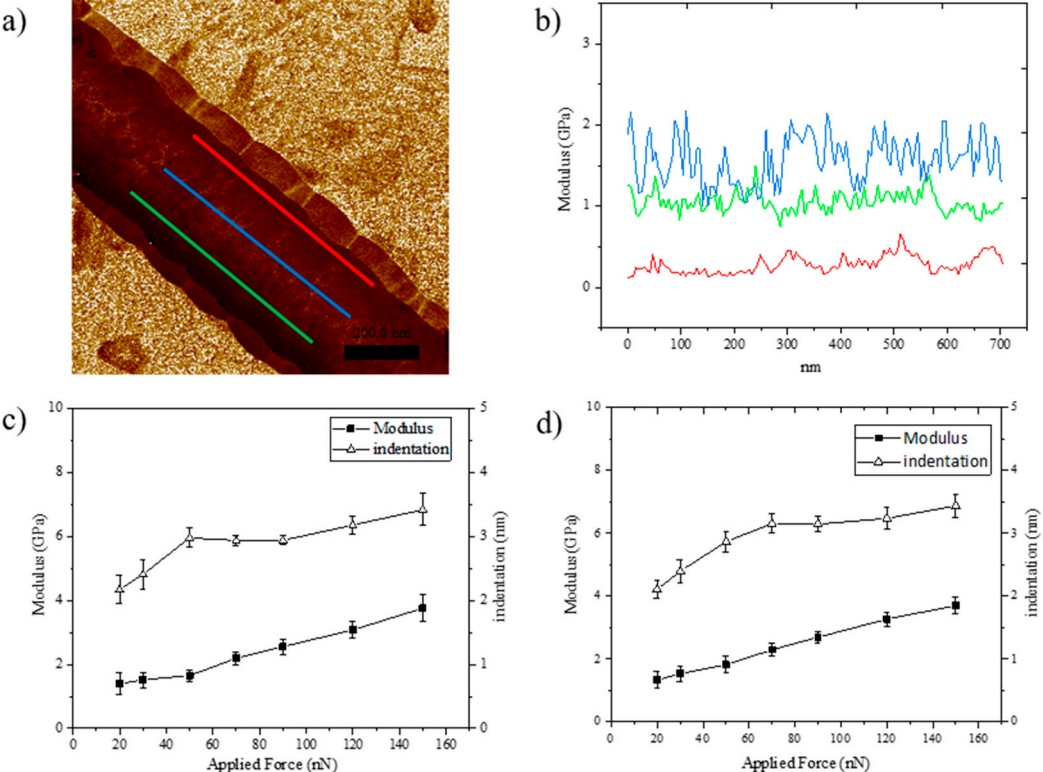

**Figure 4.** (**a**) Modulus image of an annealed nanofiber with cross-section lines at three different locations along the fiber axis (peakforce setpoint = 30 nN). (**b**) Modulus along those three lines. Evolution of modulus and indentation depth with respect to applied peakforce of (**c**) pristine and (**d**) annealed nanofibers.

It should be noted that two precautions should be taken for an accurate study of the mechanical properties of the nanofibers. First, the modulus value should be measured along the spine of a nanofiber to minimize the effect of the tip convolution. Figure 4a,b illustrate the difference in modulus between three cross-section lines, one was taken at the top of the nanofiber and the other two were taken closer to the edge. It is clear that the modulus values taken at the top are always higher than those taken on the sides of the nanofiber. Due to the curvature of the fiber, the tip cannot not fully touch its sides which results in different contact areas as well as applied stress. This curvature effect is less pronounced at the highest points (the spine) of the nanofibers.

Secondly, the modulus in peakforce QNM is estimated assuming that a pure elastic response of the material to the applied force has occurred. Therefore, an investigation of the material behavior with respect to applied force should be carried out prior to the main study. Figure 4c,d display the evolution of the indentation depth and the contact modulus as a function of the applied force for representatives as-spun and annealed nanofibers with similar diameters. For as-spun (amorphous) nanofiber, the indentation depth gradually increases while the contact modulus values remain stable when the peakforce values are in the range from 20 nN to 50 nN. This behavior of the contact modulus,

in which it is independent of the applied force, is mostly observed in elastic materials. When the peakforce continues to increase from 50 nN to 150 nN, however, the responses are different than lower values. The indentation depth remains unchanged and then increases while the contact modulus values keep increasing with respect to the increase of applied force. This means that the responses are no longer elastic. Hence, we assume that plastic deformation has occurred when the peakforce is larger than 50 nN. As for crystalline nanofiber, the contact modulus values are stable while the indentation depth increases when the peakforce setpoint is lower than 70 nN. When the force becomes larger, plastic deformation starts to take place. As expected, the indentation depth gradually increases for both samples when the peak force increases from 20 nN to 150 nN. The evidence of plastic deformation can be observed by looking at the change in morphology of the nanofiber with respect to applied force (Figure S4 of Supporting Information). It should also be worth mentioning that the same trend was observed for all nanofibers regardless of their diameters. This means that the behavior does come from the sample response without any contribution from the underlying substrates. From this analysis, we have chosen a value of 30 nN for the peakforce which is within the elastic regime of both kinds of samples.

### 3.4.2. Data Processing and Relevant Model

For data analysis, the traditional way to extract contact modulus values requires determination of a contact point, which can be used for calculating the deformation as well as fitting the data (the contact point-based method). Determining the contact point is usually straightforward in the case of a rigid, non-deformable sample but becomes more difficult for soft, adhesive samples (e.g., polymeric materials). For example, there is no sharp contact in both cases of PLLA nanofibers, instead, we observe a force distance curve with a rounded bottom (compared to straight lines for more rigid samples) indicating that most probably the tip was gliding onto the sample surface. In addition, the signature of the adhesion force in the unloading/retraction part of the force curve also makes it more complicated to evaluate the contact point (Figure S5 of Supporting Information). Therefore, the most reliable method, in this case, is the linearized model which does not require the assignment of the contact point for the modulus calculation. For example, the linearized equation for the DMT model to fit the slope of the unloading curves can be described as:

$$(F)^{2/3} = \left( \frac{4}{3} \cdot \frac{E}{(1 - \nu^2)} \cdot \sqrt{R_{Eff}} \right)^{2/3} \cdot \delta \tag{2}$$

where $F$ is the total applied force to the tip: $F = F_{peakforce} + F_{adh}$ (nN), $E$ is the fitted DMT's modulus of the sample (GPa), $R_{Eff}$ is the effective tip radius (nm), $\nu$ is the sample's Poisson ratio and $\delta$ is the sample deformation (nm). In this study, $\nu$ is taken as 0.3 for both pristine and thermal-treated samples. $F_{adh}$ and $\delta$ are experiment-dependent parameters that can be measured in real-time simultaneously with the topography.

Furthermore, the choice of which relevant contact mechanic models (Hertz, Sneddon, DMT, or JKR) should be applied is also of great importance. The first model, Sneddon, can only be applied to cases in which the tip has a conical shape and when the indentation depth is large. In the case of PLLA nanofibers, the tip had a spherical shape with a radius of 29 nm while the indentation depths were around 2–4 nm. Therefore, the Sneddon model cannot be used. It is also clear that the Hertz model should be ruled out due to the presence of significant adhesion response in the force curves. The selection between DMT and JKR models is, however, more delicate. Maugis has proven that the Hertz, DMT and JKR models are all special cases of a more generalized contact mechanics model [48]. DMT only considers the long-range surface forces outside of the contact area while JKR assumes that adhesion forces are actually short-range surface forces within the contact area. In order to know if

whether DMT is appropriate for the PLLA nanofibers system, we estimate the Maugis parameter, $\lambda$, which can be written as [49]:

$$\lambda = \frac{2.06}{D_0} \cdot \sqrt[3]{\frac{R_{Eff} W_{adh}^2}{\pi E^2}} \tag{3}$$

where $D_0$ is the interatomic spacing describing the system, which is often taken as 0.165 nm. $W_{adh}$ is the tip–sample adhesion energy per unit area: $W_{adh} = F_{adh}/2\pi R_{Eff}$ and $E$ is the Young modulus of the tip–sample system. If $\lambda$ approaches 0, the DMT model is a good fit whereas if $\lambda$ approaches infinity, the JKR model should be used. Based on the Equation (3), the value of the Maugis parameter for the as-spun and annealed samples are found to be in the range of 2–5, which indicates that the DMT model is indeed a suitable choice to describe the system in our case.

### 3.4.3. Evaluation of the Viscoelastic Contribution

Apart from the plastic deformation, there is another factor that hampers obtaining the true elastic modulus of a sample, i.e., the effect of viscoelasticity [50]. Such viscous response, in many cases, dominates over the elastic response and should be identified separately if we want to obtain a meaningful contact modulus value from the measurements. Unfortunately, a technique like peakforce QNM normally involves applying a contact mechanic model to the obtained quasi-static force–distance curves. Such models are only applicable under the assumption that the tip–sample interaction is linearly dependent on the cantilever deflection and, therefore, those models do not take into account the dependence on the tip velocity or, in other words, the viscous response. Therefore, we employed ImAFM which is a dynamic, multi-frequency, AFM-based measurement method. In this technique, the tip is driven and oscillated at two different frequencies that are close to the resonant frequency of the cantilever. The tip–surface interaction results in an intermodulation spectrum (a mixing product of frequencies) from which, by Fourier transformation, one can generate the dynamic force quadratures $F_I$ and $F_Q$ at each pixel. These force quadratures do not display force as a function of tip position but represent integrals of the force over a single oscillation cycle. The $F_I$ curve shows the integrated force that is in phase with the cantilever motion and can be described as conservative force whereas $F_Q$ shows the integrated force that is in phase with the velocity and is attributed to the dissipative force or energy lost during each tapping cycle of the cantilever. The $F_I$ and $F_Q$ can be considered as force curves in dynamic AFM and can be used to explain various tip–surface interactions.

To validate the efficiency of the method, we first characterize a reference sample which is a polymer blend of poly(styrene) (PS) and low-density poly(ethylene) (LDPE) using the same tip that was used for PFQNM measurements. PS acts as the stiff matrix with a modulus of 2.5–2.8 GPa whereas LDPE acts as the soft viscous filler with an elastic modulus around 0.3 GPa. The ImAFM results of the polymer blends are shown in Figure S6 of the Supporting Information. The $F_I$ and $F_Q$ curves on PS and LDPE regions are completely different. The $F_I$ value of the PS region displays a large negative value at large amplitude, meaning that the force is fundamentally repulsive and the $F_Q$ value is significantly smaller. On the contrary, the $F_I$ curve of LDPE shows positive values corresponding to a dominantly attractive force, even at maximum amplitude. Its $F_Q$ curve also displays a sizable negative value. Finally, for the stiff and mostly elastic response of PS region, we can fit the DMT model to the slopes of $F_I$ (A) to generate a map of the modulus from which we find a value for the PS region is about 2.5 GPa, i.e., fully consistent with the macroscopic measurements.

After successfully calibrating with the reference sample, we analyzed the PLLA nanofibers with ImAFM. Figure 5 shows the $F_I$ and $F_Q$ curves of both as-prepared and annealed electrospun fibers. At small oscillation amplitudes, the tip is far away from the surface resulting in the zero values for both $F_I$ and $F_Q$. With the gradual increase of oscillating amplitude, the tip starts to touch and penetrate to the surface sample, reaching the largest value of $F_I$ and $F_Q$ at the maximum amplitude. It is clear that, in both cases, the $F_I$ shows large negative values while smaller values are observed for $F_Q$. It is important to note the absence of any hysteresis loop in the $F_I$ curves of both as-spun and annealed

electrospun fibers meaning that their surface is completely recovered after each cycle, or in other words, a dominantly elastic response.

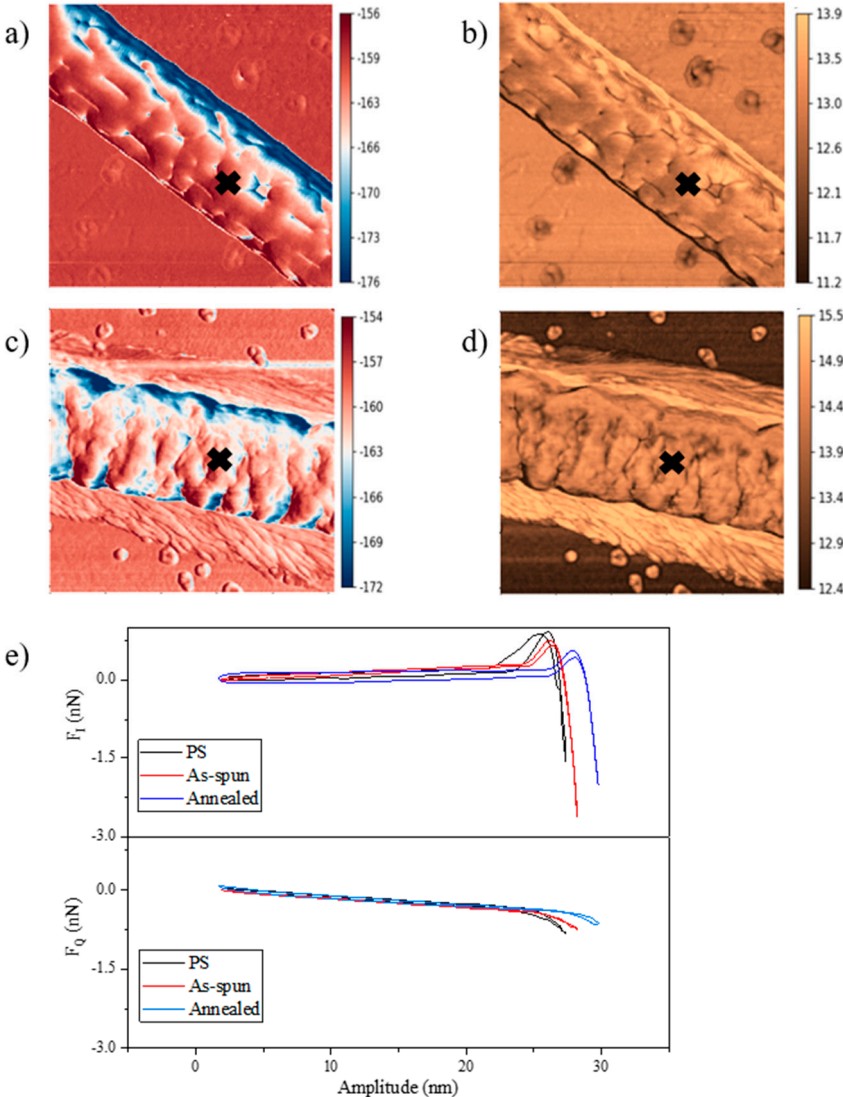

**Figure 5.** ImAFM phase (°), amplitude (nm) of as-spun (**a**,**b**) and annealed (**c**,**d**) PLLA nanofibers, respectively; (**e**) $F_I$ and $F_Q$ curves of those samples taken at the black crosses. The black $F_I$ and $F_Q$ curves correspond to one pixel from a PS domain of the reference sample (see Supporting Information).

### 3.4.4. Comparison of the Elastic Modulus between Pristine and Annealed Nanofibers

The overlapping of the loading and unloading curves in PFQNM and the absence of hysteresis in the $F_I$ curves in ImAFM are two pieces of evidence proving that there is neither plastic deformation nor viscoelastic behavior in both as-spun and annealed PLLA nanofibers. Furthermore, the calculation of Maugis parameters leads to the conclusion that the DMT model is the best fit model for both materials. With all these considerations, we can thus apply the DMT model to extract the modulus value. Figure 6 shows the elastic modulus as a function of the fiber diameter for each case. It appears that the smaller the diameter of a nanofiber is, the higher the elastic modulus it has. This size effect on the elastic modulus of nanofibers has been reported in numerous studies. It has been proved that orientation in crystalline and amorphous domains are higher in thinner fibers, resulting in an increase of elastic modulus [51–53]. Since tapping mode AFM measurements showed that there is a reduction of about 30% in diameter of nanofibers from the annealing process (from 210 ± 88 nm for the pristine fibers to

160 ± 44 nm for the annealed samples), for a meaningful comparison, we should compare the elastic modulus of annealed nanofibers to that of as-spun nanofibers with a relatively higher diameter. For example, we should compare the elastic modulus of a 200-nm pristine fiber with a 170-nm annealed fiber, and so on. In this regard, the nanomechanical properties of the annealed fibers are definitely higher than those of untreated ones. The increase in elastic modulus of annealed nanofibers can be explained by the increase in the crystallinity upon annealing. The surface of annealed nanofibers consists of many highly packed, granular domains which, in turn, enhance the surface stiffness of the sample under applied stress normal to the fiber axis. It should also be noted that the elastic modulus measured by PFQNM is in the direction perpendicular to the fiber axis so these values should be expected to be different from values measured by classical methods, such as the three-point bending test, which returns the axial modulus values of the fiber.

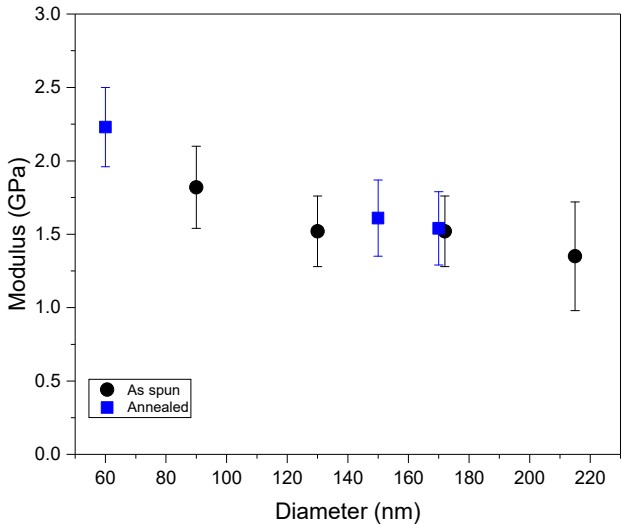

**Figure 6.** Elastic modulus as a function of the fiber diameters for as-spun and annealed nanofibers.

However, in our study, we are more interested in how the nanofibers respond to stress normal to the fiber axis because it also corresponds to the piezoelectric axis. In this case, corresponding to higher elastic modulus in the normal axis, annealed nanofibers can withstand higher stress and higher deformation, which links directly to higher energy output for a piezoelectric energy-harvesting device.

### 3.5. Local Piezoelectric Responses

The local electroactive behaviors of both pristine and annealed nanofibers were explored by employing the spectroscopic tool of the PFM. A conductive tip was used as a top electrode and scanned over the surface to assess the polarization switching of individual nanofibers. At the beginning of the experiment, tapping mode was used to explore the samples and to avoid damage caused by strong tip–sample interaction in contact mode. When a nanofiber of interest was located, the whole system was switched to the PFM interface. The PFM hysteresis loop measurements were performed in remnant mode in which pulsed, triangular DC waveforms are applied to the samples. In the "write" state (or "on-field" state), AC and DC voltage biases are applied simultaneously to the sample and both piezoelectric and electrostatic forces contribute to the final output signals. The "read" (or "off-field" state) operation was performed at zero DC voltage, thus, charge accumulation on the analyzed sample was avoided and any electrostatic contribution was suppressed (see Figure S7 of Supporting Information). For our experiments, the typical driving AC voltages were 7 V and 5 V for as-spun and annealed nanofibers, respectively. Such voltages were required to excite the PLLA fibers and therefore to be able to measure PFM activity.

Figure 7 demonstrates the out-of-plane electromechanical response of as-spun and annealed nanofibers as a function of the DC bias voltages. Butterfly-shaped responses in amplitudes, which are typical for piezoelectric materials, can be clearly observed in both as-spun and annealed nanofibers. Furthermore, the intrinsic ferroelectric behavior of both samples is also proved by the presence of a strong hysteresis in the phase signal associated to the 180° phase difference indicating the presence of two stable states with opposite polarity. Amorphous PLLA electrospun nanofibers also behave as amorphous piezoelectric polymers [54]. The origin of the ferroelectricity comes from the presence of oriented permanent dipoles. Indeed, the stretching force that occurs in the electrospinning process aligns all the C=O dipoles and creates a net polarization perpendicular to the fiber axis.

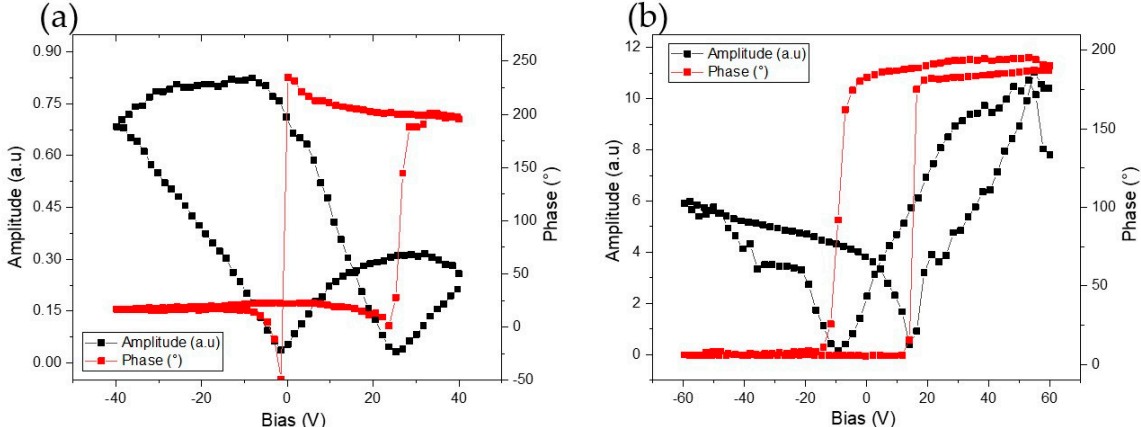

**Figure 7.** Out-of-plane PFM amplitude and phase hysteresis loops of (**a**) as-spun and (**b**) annealed PLLA nanofibers. All measurements were recorded in remnant mode to avoid electrostatic contributions.

As mentioned in the introduction section, nanoscale longitudinal piezoelectric responses for PLLA nanofibers were already recorded by PFM measurements. Sultana et al. [34] have observed similar electroactive response in flexible and wearable device based on the amorphous PLLA nanofiber membrane. We note the PFM loops recorded on the as-spun fibers are shifted towards the positive voltage values, pointing out an imprint phenomenon in the local switching properties, commonly attributed to an internal built-in electric field which exists in the piezoelectric material [55,56].

From the plotted loops, although no quantitative value for deformation amplitude detected by the PFM tip is available, clear enhancement in local piezo activity is observed when probing annealed nanofiber (Figure 7b). This result is significant considering lower driving voltage applied to this thermal-treated nanofiber (5 V) compared to the higher AC voltage (7 V) applied to the as-spun nanofibers. This improved electromechanical behavior can be explained by the fact that the crystallinity in the heat-treated sample was much higher. The occurrence of a longitudinal piezoelectric coefficient ($d_{33}$) in crystalline PLLA electrospun nanofibers agrees with the fact that the α′-phase of PLLA have a space group P12$_1$1 [57], and therefore, belong to crystal class (C2). For materials of this class, the $d_{33}$ exists [57]. Therefore, the qualitative PFM results revealed the promising potential of both amorphous and thermally treated PLLA nanofibers in terms of energy harvesting applications.

## 4. Conclusions

The effect of an annealing process on the crystallinity of PLLA electrospun nanofibers was studied by DSC, FTIR, and XRD techniques while the enhancement in nanomechanical and piezoelectric properties was revealed by AFM-based methods. The annealing process strongly increases the crystallinity of the nanofibers, from dominantly amorphous into a material with 50% of crystallinity. The investigation by ImAFM revealed that, for PLLA nanofibers, elastic response was dominant over the viscous response and the DMT model is then an appropriate model to extract the values of the contact modulus proven by the fact that the Maugis parameter λ remains small. As shown

by both PFQNM and ImAFM results, annealed nanofibers possess an improved modulus in the direction normal to fiber axis. Both amorphous and annealed semi-crystalline nanofibers present a ferroelectric and piezoelectric response at the local scale. Because of their higher crystallinity, the annealed nanofibers also show better out-of-plane PFM responses which links to a better piezoelectric property. In conclusion, by enhancing the piezoelectricity and mechanical properties of the nanofibers, the post-annealing process could play an important role for PLLA electrospun fibers in the context of energy harvesting applications, such as self-powered wearable devices.

**Supplementary Materials:** The following are available online at http://www.mdpi.com/2076-3417/10/2/652/s1, Figure S1: Tapping AFM height images of several pristine PLLA nanofibers. Figure S2: Tapping AFM height images of several annealed PLLA nanofibers. Figure S3: Tapping AFM height images of large area of as-spun nanofibers. Figure S4: peakforce QNM height images corresponding to different applied forces; Figure S5: Typical force–separation curve obtained by peakforce QNM. Figure S6: ImAFM results of PS-LDPE reference sample. Figure S7: Triangular voltage bias signals applied to the nanofibers.

**Author Contributions:** Conceptualization of the experimental research, P.L. and S.B.; TMAM, PFQNM, ImAFM experiments and discussions, N.T.C. and P.L.; Electrospinning, N.T.C., M.D., and N.T.; FTIR and XRD analysis, S.B., M.D. and N.T.; PFM experiments, A.D.C. and A.F. All authors discussed the results and implications and commented on the manuscript at all stages, R.L., G.-M.R. All authors have read and agreed to the published version of the manuscript.

**Acknowledgments:** This work was done under the framework of BIOHARV (BIObased Energy HARvesting Solutions Using poly (L-Lactide) INTERREG V FWVL Project. Scanning probe research activities in Mons were supported by the FRS-FNRS PDR Project "Hybrid Organic/Inorganic Nanomaterials for Energy COnversion and STOrage Devices on FLEXible and Stretchable Substrates" (ECOSTOFLEX). The authors thank the "Région Hauts-de-France", the "Fonds Européen de Développement Régional" (FEDER) and the Chevreul Institute (FR 2638) for funding the X-ray diffusion/diffraction equipment and the electrospinning system. Chevreul Institute is supported by the "Ministère de l'Enseignement Supérieur, de la Recherche et de l'innovation", the "Région Nord-Pas de-Calais" and the "Fonds Européen de Développement des Régions". Région Hauts-de-France and Fonds Européen de Développement Régional (FEDER) are also gratefully acknowledged for funding the MFP-3D microscope under the Program, "Chemistry and Materials for a Sustainable Growth". The authors thanks Jean-Francois Tahon for the WAXS experiments. P.L. and J.-M.R. are Senior Research Associates of the FRS–FNRS and Research Associate of the FRS–FNRS (Belgium), respectively.

**Conflicts of Interest:** The authors declare no conflict of interest.

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
