# Peer review of "On the Nanoscale Mapping of the Mechanical and Piezoelectric Properties of Poly (L-Lactic Acid) Electrospun Nanofibers"

_applsci, doi:10.3390/app10020652_

Round 1

Reviewer 1 Report

The manuscript by Leclere and coworkers describes the effects of thermal annealing on the mechanical and piezoelectric properties of PLLA nanofibers. The main conclusion states that the as-spun nanofibers have low crystallinity and that thermal annealing significantly increases the crystalline content of the fibers.

The increase in crystallinity is well characterized by DSC, FTIR, and XRD. However, while the nanomechanical properties measured by AFM-based methods are discussed in-depth (different models and the interpretation of the data), the measured elastic moduli reported are those in the normal axis and not the axial axis of the fibers. The authors mentioned that the response to stress along the axis are more relevant, but provided no such measurement.

I will only recommend this manuscript for publication after a major revision to address the following:

Increase in crystallinity of semi-crystalline polymers from thermal annealing is well published in literature. The authors should discuss whether how this effect is similar or different in bulk polymer, thin films, and nanofibers. A large section of the manuscript discusses the measurement of elastic moduli using AFM-based method with the conclusion that the measured normal axis moduli are inconsequential to the performance of a nanofiber. The authors should revise this section to include the measurement of the axial moduli. The discussion that accompanied Figure 6 regarding the effect of diameter on elastic moduli can be expanded upon. Why do fibers with the same diameter, as-spun and annealed, have the same modulus? Does this result imply that the size of the fiber is more indicative of the normal-axis moduli? The manuscript requires a thorough check for spelling errors. For example, “DSC” in the abstract is incorrectly written. While the message is clear, these small mistakes do distract the readers.

Author Response

Comment from reviewer 1:

The increase in crystallinity is well characterized by DSC, FTIR and XRD. However, while the nanomechanical properties measured by AFM-based methods are discussed in-depth (different models and the interpretation of the data), the measured elastic moduli reported are those in the normal axis and not the axial axis of the fibers. The authors mentioned that the response to stress along the axis are more relevant, but provided no such measurement.

Answer:

We thank the referee for his comment. In the manuscript, we wrote that “However, in our study, we are more interested in how the nanofibers respond to stress along the axis because it also corresponds to the piezoelectric axis”. The initial intention of using the word “along” in this case is to indicate the direction in which the data points were taken. However, we admit that this word is indeed ambiguous and could create misunderstanding. The modulus values measured by PeakForce QNM were in the direction normal to the fiber axis and were different from axial modulus value. Therefore, we will change the above sentence to: “However, in our study, we are more interested in how the nanofibers respond to stress normal to the fiber axis because it also corresponds to the piezoelectric axis”.

Increase in crystallinity of semi-crystalline polymers from thermal annealing is well published in literature. The authors should discuss whether how this effect is similar or different in bulk polymer, thin films, and nanofibers.

Answer:

We are agreeing with the referee about the other possible forms of PLLA materials. Since our focus here is on electrospun nanofibers we didn’t put the values for bulk properties. We also processed PLLA films (tens of µm in thickness) by compression molding (not mentioned in this manuscript). The films were then post-annealed to enhance the crystallinity following a similar procedure as we did with PLLA nanofibers. As indicated by DSC characterization (see attached pdf file), the as-processed PLLA films were mostly amorphous whereas the annealed PLLA films show significant improvement of crystallinity.

The non-treated PLLA film shows an exothermic peak related to cold crystallization at 113.8 whereas the relative peak disappears in the annealed film. Therefore, the effect of annealing process on the crystallinity of bulk PLLA films is expected to be like that on PLLA nanofibers. Even if the quantitative values are different in two cases, the trend is the same for the two forms (increasing of the crystallinity)

Sample

Tc ()

Tm ()

Xc (%)

Nanofiber (untreated)

85.6

173.7

21

Nanofiber (annealed)

-

173.6

501

Film (untreated)

113.8

174.5

11

Film (annealed)

-

173.0

611

A large section of the manuscript discusses the measurement of elastic moduli using AFM-based method with the conclusion that the measured normal axis moduli are inconsequential to the performance of a nanofiber. The authors should revise this section to include the measurement of the axial moduli. The discussion that accompanied Figure 6 regarding the effect of diameter on elastic moduli can be expanded upon. Answer:

We thank the referee for raising this important question. Technically speaking with the AFM set-up (Dimension Icon system) in PeakForce Tapping mode)) we can only measure perpendicularly to the fiber axis. We also well aware that in literature other methods (3-point bending for instance) are able to provide the axis modulus but here since the most pertinent axis for the piezoelectricity is perpendicular to the fiber axis (d33), we do not need to consider this approach.

Why do fibers with the same diameter, as-spun and annealed, have the same modulus? Does this result imply that the size of the fiber is more indicative of the normal-axis moduli?

We were also surprised by the observation but we believe that because there is a large amount of amorphous domains existing in the surface of the nanofibers even after annealing. The crystalline domains were possibly wrapped by some amorphous thin envelope. Therefore, the measured modulus values are most probably the average value of crystalline and amorphous fractions and with the sample, the rigid crystalline parts could slightly move together with the tip during the tip/sample interaction.

Reviewer 2 Report

This is a very nicely presented paper in an area of interest to researchers in PLLA. There are some minor errors in language which need to be corrected and it is important that the error in the title (the extra words "at the") be removed. Similarly saying DCS rather than DSC in the Abstract will result in search engines not properly classifying the paper. 

The following points are noted for action:

1. The scales in some of the micrographs are either hard to read or absent: Figures 4(a), S1 and S2.

2. The Tg for the as-spun nanofibers is apparent in Figure 1 but not for the annealed fibers due to the reduction in the amorphous fraction. It would have been useful in Figure 1 to expand the traces in the region of Tg to show the enthalpy change and thus whether there are any changes on annealing.

3. Reference 13 has an incorrect citation of the authors and is also incomplete. Please give the correct authors and the doi link since this an open source document. In relation to the value for the enthalpy value cited in reference 13, this is not a primary DSC reference and a better citation would be Righetti et al European Polymer Journal 2015, 70, 215. This will have a minor effect on the values of degree of crystallinity quoted.

Author Response

Comments from reviewer 2:

The scales in some of the micrographs are either hard to read or absent: Figures 4(a), S1 and S2

We modified the figures by adding a scale bar.

The Tg for the as-spun nanofibers is apparent in Figure 1 but not for the annealed fibers due to the reduction in the amorphous fraction. It would have been useful in Figure 1 to expand the traces in the region of Tg to show the enthalpy change and thus whether there are any changes on annealing.

We agree with the comment from the referee and the figure (see pdf attached) shows the enlarged of DSC diagram for as spun and annealed PLLA nanofibers. There’s a small shift in the glass transition temperature (Tg).

Reference 13 has an incorrect citation of the authors and is also incomplete. Please give the correct authors and the doi link since this is an open source document. In relation to the value for the enthalpy value cited in reference 13, this is not a primary DSC reference and a better citation would be Righetti el al European Polymer Journal 2015, 70, 215. This will have a minor effect on the values of degree of crystallinity quoted.

We thank the referee for his comment.

We modify the ref 13 accordingly and add a more appropriate reference for the enthalpy issue.

Reference [13] will be corrected into: “Carlos, A.; Olivas-Armendariz, I.; Castro-Carmona, J.S. Scaffolds for Tissue Engineering Via Thermally induced Phase Separation. Adv. Regen. Med. InTech. 2011. http://dx.doi.org/10. 5772/25476”

Relative to the enthalpy of melting for PLA single crystal, the reference will be changed to: “Righetti, M.S.; Gazzano, M.; Laura Di Lorenzo, M. Enthalpy of melting of α’- and α- crystal of poly(L-lactic acid). European Polymer Journal. 2015, 70, 215-220.”

And the indicative number of reference will be changed from [13] to [35].

We, of course, updated the reference numbering.

Round 2

Reviewer 1 Report

The authors addressed my questions appropriately. I support the publication of this manuscript.